# VERY LARGE SCALE MULTI-AGENT REINFORCEMENT LEARNING WITH GRAPH ATTENTION MEAN FIELD

## ABSTRACT

With recent advances in reinforcement learning, we have witnessed countless successes of intelligent agents in various domains. Especially, multi-agent reinforcement learning (MARL) is suitable for many real-world scenarios and has vast potential applications. However, typical MARL methods can only handle tens of agents, leaving scenarios with up to hundreds or even thousands of agents almost unexplored. There exist two key challenges in scaling up the number of agents: (1) agent-agent interactions are critical in multi-agent systems while the number of interactions grows quadratically with the number of agents, causing great computational complexity and difficulty in strategies-learning; (2) the strengths of interactions vary among agents and over time, making it difficult to precisely model such interactions. In this paper, we propose the Graph Attention Mean Field (GAT-MF) method, where we convert agent-agent interactions into interactions between each agent and a weighted mean field, greatly reducing the computational complexity. We mathematically prove the correctness of this conversion. We design a graph attention mechanism to automatically capture the different and time-varying strengths of interactions, ensuring the ability of our method to precisely model interactions among the agents. We conduct extensive experiments in both manual and real-world scenarios with up to more than 3000 agents, demonstrating that comparing existing MARL methods, our method reaches superior performance and 9.4 times computational efficiency.

## 1 INTRODUCTION

In recent years, rapid progress in reinforcement learning (RL) has largely facilitated humans' decision-making in complex situations. In various domains such as game playing (Silver et al., 2017; Ye et al., 2021), robotics (Sinha et al., 2022; Brunke et al., 2022), public health (Bastani et al., 2021; Hao et al., 2021; 2022), and even nuclear fusion system (Degrave et al., 2022), RL algorithms and applications keep emerging. Most of the successful RL works focus on single-agent scenarios, while in the real world, it is normal that a system consists of multiple agents, and interactions among the agents are of vital importance. Therefore, multi-agent reinforcement learning (MARL) has especially wide applications and developments in corresponding methods are called for.

In fact, previous researchers have done plentiful works on MARL. For example, MADDPG (Lowe et al., 2017) outperforms single-agent DDPG in experiments with various goals, a series of studies such as QMIX (Rashid et al., 2018), AlphaStar (Vinyals et al., 2019) and MAPPO (Yu et al., 2021) keep surpassing human professional players and refreshing the score on the Starcraft game, which is a typical MARL benchmark. Also, there are various social and industrial applications of MARL methods in traffic signal control (Wang et al., 2021b), power distribution management (Wang et al., 2021a), cloud computing (Balla et al., 2021), etc. However, existing studies typically only consider no more than tens of agents, while methods dealing with scenarios with up to hundreds or even thousands of agents remain almost unexplored.

Despite scenarios with hundreds or thousands of agents are common in the real world, there exist two key challenges in scaling up the number of agents in MARL methods. (1) **Large number of agent-agent interactions**. In multi-agent systems, the agents naturally interact with each other all the time, and such interactions are critical for the system dynamic. Therefore, besides simple agent-environment interactions, MARL methods must take the agent-agent interactions into con-

sideration to reach good performance. However, the number of agent-agent interactions increases quadratically following $O(N^2)$ as the number of agents grows to $N$, which greatly adds to the computational complexity and difficulty for the agents to learn efficient strategies. (2) **Varying strengths of agent-agent interactions**. Due to the intrinsic dynamism of real-world scenarios, the strengths of interactions not only vary among each pair of agents but also vary over time, making it difficult to precisely model all the agent-agent interactions. If we manually set the interaction strength of each pair of agents according to prior knowledge of a certain scenario, it requires repeated manual work when we want to train models to solve problems in different scenarios. On the other hand, it is almost impossible to do such manual work when the number of agents is large.

In view of these challenges, we propose Graph Attention Mean Field (GAT-MF) method to largely scale up the number of agents in MARL. First, to solve the problem of the unaffordably large number of agent-agent interactions, we develop the previous study of unweighted Mean Field (Yang et al., 2018) into a weighted one. We prove the mathematical correctness of converting interactions among the agents into the interactions between each agent and a corresponding field, which is obtained through a weighted average over the raw agent-agent interactions. By such conversion, the number of agent-field interactions only increases linearly following $O(N)$ with $N$ agents, which greatly reduces the computational complexity and difficulty in learning efficient strategies. Also, such conversion keeps the information of different interaction strengths among agents in the weights, which is discarded in the unweighted mean field. Second, to automatically capture the varying strengths of interactions, i.e., the weights in calculating the equivalent field, we model the relations among the agents into a graph where each node represents one agent. We design a graph attention mechanism to dynamically learn and calculate the different and time-varying interaction strengths among the agents, which requires neither prior knowledge of the system nor manual work. Third, we evaluate our GAT-MF method in (1) a grid-world manual scenario with 100 agents (see Section 5) and (2) a real-world metropolitan scenario with more than 3000 agents, which is built according to real-world data (see Section 6). The results show that the proposed method outperforms the existing MARL methods in both scenarios and prove its ability in scaling up to scenarios with a large number of agents. Besides, the results also indicate that our method reaches high computational efficiency, taking only 41.5% GPU memory and reaching up to 9.4 times training speed.

In summary, the main contributions of this work include:

- We prove the mathematical correctness of converting agent-agent interactions in multi-agent systems into the interactions between the agent and a weighted mean field. By doing so, we greatly reduce the computational complexity and difficulty in learning efficient strategies and make it possible to scale up to scenarios with a very large number of agents.

- We design a graph attention mechanism to automatically capture the varying strengths of the agent-agent interactions, ensuring that our method can precisely model these interactions without prior knowledge of the strengths of the interactions in the scenario.

- We conduct extensive experiments in both a manual grid-world scenario and a real-world metropolitan scenario with up to more than 3000 agents. The results demonstrate that comparing the typical MARL methods, our method achieves superior performance in both scenarios and obtains 9.4 times computational efficiency.

## 2 RELATED WORKS

**Large Scale Task with RL.** Many previous studies focus on solving large-scale tasks with RL. One common approach is to aggregate the large number of natural units into a relatively small number of clusters and control each cluster with one agent (Wang et al., 2021b; Qiu et al., 2021; Hao et al., 2022). Another kind of methods decompose the raw vast action space into a hierarchical one according to prior knowledge of the targeted scenario, simplifying the decision-making process (Hao et al., 2021; Ma et al., 2021; Ren et al., 2021). Other researchers combine human experts' solutions to aid the RL agent to learn more efficient strategies in large-scale scenarios (Qu et al., 2019; Li et al., 2022; Hao et al., 2022). Although these works achieve success in their targeted scenarios, the manual techniques of unit aggregation, action space decomposition, or experts' solution collection are largely problem specific and require strong prior knowledge of the scenarios. In contrast, our method provides a direct MARL approach, requiring neither manual techniques nor prior knowledge, and thus is able to easily train models for solving problems in different scenarios.

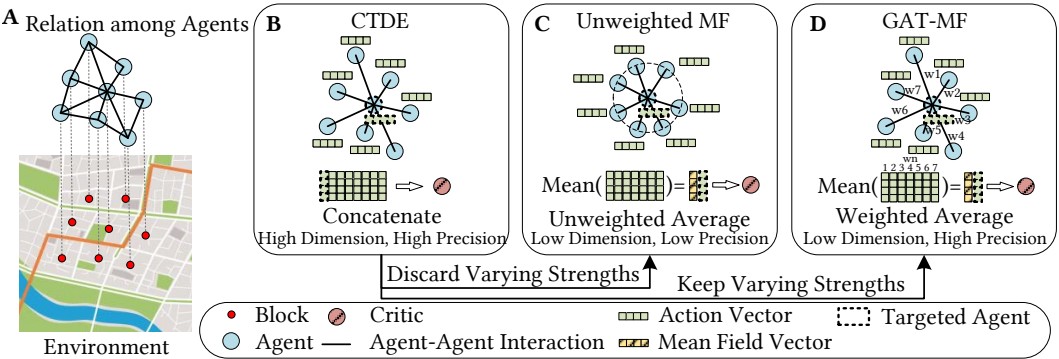

Figure 1: Comparison among existing MARL methods and our proposed method. **(A)** Example of a multi-agent scenario in a metropolitan where each block corresponds to one agent and the relations among adjacent blocks lead to agent-agent interactions. **(B)** Limitation of the CTDE methods, i.e. high dimension of the concatenated input vector of the critic. **(C)** Limitation of the unweighted MF, i.e., losing precision on modeling agent-agent interactions using unweighted average. **(D)** Key design of our proposed GAT-MF method, reducing the dimension while keeping the precision.

**Multi-agent Reinforcement Learning (MARL).** There exist abundant works on MARL and here we illustrate the differences among various MARL methods with an example in a metropolitan in Figure 1. The main approaches of recently popular MARL solutions can be roughly summarized into two categories, one is value decomposition (VD) and the other is centralized training with decentralized execution (CTDE). VD methods factorize the joint Q-function into a certain function of the local Q-function of each agent in order to reduce the complexity. VDN (Sunehag et al., 2017) uses an additive function, QMIX chooses a monotonic function (Rashid et al., 2018) and QTRAN extends it into a more general function (Son et al., 2019). However, it is not easy to factorize the joint Q-function into hundreds or even thousands of terms, limiting the scalability of such methods on a large number of agents. On the other hand, CTDE methods such as MADDPG (Lowe et al., 2017), COMA (Foerster et al., 2018) and MAPPO (Yu et al., 2021) design decentralized actor and centralized critic (or value function), where the latter one only works during the training process. Since the centralized critic (or value function) requires concatenating the vectors of local states and actions from all agents together as its input to obtain the information on agent-agent interactions, it is obvious that when the number of agents is large, the concatenated vector will be extremely high in dimension, making the training process hard (Figure 1B). In contrast, due to our special design, our proposed method can scale up to scenarios with more than 3000 agents and keeps high computational efficiency at the same time.

**Mean Field (MF) MARL.** In order to reduce the unaffordable high vector dimension in CTDE methods mentioned above and scale up the number of agents, a special technique named Mean Field (MF) (Yang et al., 2018) is proposed. It approximates the concatenation of actions with the unweighted average of these actions, namely mean field. RL algorithms with MF technique have been successfully applied in domains such as ridesharing order dispatching (Li et al., 2019), mobile networks management (Shi et al., 2020) and unmanned aerial vehicles (UAVs) control (Chen et al., 2020). Nevertheless, such approximation regards interactions among all agents equally and discards the varying strengths of these interactions, losing precision in modeling the complex relations among the agents (Figure 1C). In this paper, we develop the MF method into GAT-MF, in which we design a variant of graph attention mechanism to automatically capture the varying strengths of agent-agent interactions and calculate a weighted average of the actions according to their strengths. Therefore, we are able to reduce the vector dimensions, scale up the number of agents, and keep precise modeling of the relations among the agents at the same time (Figure 1D).

## 3 PRELIMINARIES

**Markov Decision Processes (MDPs).** In this paper, we mainly discuss our method considering a multi-agent version of Markov Decision Processes (MDPs), which can be defined by

$\langle n, \mathcal{S}, \rho, \mathcal{A}, P, R, \gamma \rangle$, where $n$ denotes the total number of agents. The global state $\boldsymbol{s} = (s_1, ..., s_n) \in \mathcal{S}$ consists of local state of each agent. The possibility distribution of initial state is given by $\rho = \mathcal{D}(\mathcal{S})$, where $\mathcal{D}(\mathcal{S})$ is a collection of possibility distribution over the state space $\mathcal{S}$. The joint action $\boldsymbol{a} = (a_1, ..., a_n) \in \mathcal{A}$ consists of local action of each agent and is produced by the policy $\boldsymbol{\pi_\theta} : \mathcal{S} \mapsto \mathcal{D}(\mathcal{A})$, with $\mathcal{D}(\mathcal{A})$ being a collection of possibility distribution over the state space $\mathcal{A}$. $\boldsymbol{\pi_\theta}$ is parameterized with $\boldsymbol{\theta} = (\theta_1, .., \theta_n)$ and local action of each agent is produced by $\pi_{\theta_i}(\boldsymbol{s})$. The state transition probability and the one-step reward given the current state and the joint action are defined by $P : \mathcal{S} \times \mathcal{A} \mapsto \mathcal{D}(\mathcal{S})$ and $R : \mathcal{S} \times \mathcal{A} \mapsto \mathbb{R}$ correspondingly. The global one-step reward is the sum of local one-step rewards from each agent, i.e., $R(\boldsymbol{s}, \boldsymbol{a}) = \sum_i r(s_i, a_i)$, and the long-term discounted reward from $t_0$ is defined by discount factor $\gamma$ following:

$$\boldsymbol{R_{t_0}} = \sum_{t=t_0}^{T} \gamma^{t-t_0} R(\boldsymbol{s}^t, \boldsymbol{a}^t), \tag{1}$$

where $T$ is the max length of an episode and $\gamma \in [0, 1]$.

**Q-Learning and Deep Q-Network (DQN).** Q-Learning (Watkins & Dayan, 1992) is one of the basic RL methods, which learns efficient policy in MDPs through an off-policy manner. It aims at finding a value function $Q^{\boldsymbol{\pi_\theta}}(\boldsymbol{s}, \boldsymbol{a})$ for policy $\boldsymbol{\pi_\theta}$:

$$Q^{\boldsymbol{\pi_\theta}}(\boldsymbol{s}, \boldsymbol{a}) = \mathbb{E}[\boldsymbol{R_t} | \boldsymbol{s} = \boldsymbol{s}^t, \boldsymbol{a} = \boldsymbol{a}^t] = \mathbb{E}_{\boldsymbol{s'}}[R(\boldsymbol{s}, \boldsymbol{a}) + \gamma \mathbb{E}_{\boldsymbol{a'} \sim \boldsymbol{\pi_\theta}}[Q^{\boldsymbol{\pi_\theta}}(\boldsymbol{s'}, \boldsymbol{a'})]], \tag{2}$$

where the recursive form is Bellman Equation. In practical training algorithms, this value function is obtained by minimizing the loss function which is designed with greedy thinking:

$$\mathcal{L} = \mathbb{E}_{(\boldsymbol{s}, \boldsymbol{a}, R, \boldsymbol{s'}) \sim \mathcal{B}}[(Q(\boldsymbol{s}, \boldsymbol{a}) - y)^2], \quad y = R + \gamma \max_{\boldsymbol{a'}} Q'(\boldsymbol{s'}, \boldsymbol{a'}), \tag{3}$$

where $\mathcal{B}$ is the replay buffer collecting experiences $(\boldsymbol{s}, \boldsymbol{a}, R, \boldsymbol{s'})$ from agent-environment interactions and $Q'$ is the target version of $Q$, whose parameters are synchronized from $Q$ with delay. After obtaining the optimal value function $Q^*(\boldsymbol{s}, \boldsymbol{a})$, the optimal policy $\boldsymbol{\pi}^*$ is obtained in a greedy way:

$$\boldsymbol{\pi}^*(\arg \max_{\boldsymbol{a}} Q^*(\boldsymbol{s}, \boldsymbol{a}) | \boldsymbol{s}) \to 1. \tag{4}$$

Deep Q-Network (DQN) (Mnih et al., 2013) keeps the same mathematical essence as Q-learning but approaches the optimal value function $Q^*(\boldsymbol{s}, \boldsymbol{a})$ with a deep neural network, being able to represent more complex environmental situations and thus can solve problems in more scenarios.

**Policy Gradient (PG) and Deep Deterministic Policy Gradient (DDPG) Algorithms.** Since $argmax_{\boldsymbol{a}}$ is used in obtaining the optimal policy in Q-Learning and DQN, they are only practical in scenarios where the action space $\mathcal{A}$ is discrete. In order to solve problems with continuous action spaces, Policy Gradient (PG) methods are proposed (Sutton et al., 1999). In a major group of deep PG methods, besides the value function network, the policy $\boldsymbol{\pi_\theta}$ is also approached by a neural network and directly calculates the action $\boldsymbol{a}$ given $\boldsymbol{s}$. These methods keep training the value network in a similar way as Q-Learning but without greed:

$$\mathcal{L} = \mathbb{E}_{(\boldsymbol{s}, \boldsymbol{a}, R, \boldsymbol{s'}) \sim \mathcal{B}}[(Q(\boldsymbol{s}, \boldsymbol{a}) - y)^2], \quad y = R + \gamma Q'(\boldsymbol{s'}, \boldsymbol{a'}), \tag{5}$$

while optimizing the policy network to maximize the episode return, following the gradient:

$$\nabla_{\boldsymbol{\theta}} J = \mathbb{E}_{\boldsymbol{a} \sim \boldsymbol{\pi_\theta}}[\nabla_{\boldsymbol{\theta}} \log \boldsymbol{\pi_\theta}(\boldsymbol{a} | \boldsymbol{s}) Q^{\boldsymbol{\pi_\theta}}(\boldsymbol{s}, \boldsymbol{a})]. \tag{6}$$

In practice, similar to the value network $Q$, the policy network $\boldsymbol{\pi_\theta}$ also has a target copy with delayed parameters synchronization. Deep Deterministic Policy Gradient (DDPG) (Silver et al., 2014) is a special variant of PG methods where the policy is converted from a possibility distribution over the action space to a deterministic action. In the DDPG training process, a small random disturbance is added to the deterministic action, helping the agent explore the potential action space. In this work, we mainly combine our proposed GAT-MF method with the original MADDPG algorithm (Lowe et al., 2017), which is a multi-agent version of DDPG.

## 4 METHODS

### 4.1 OVERVIEW

First, we briefly introduce the improvement of our GAT-MF methods comparing the previous unweighted MF (Yang et al., 2018). As we showed in Section 2 and Figure 1B & C, unweighted MF

uses the following approximation on the value function of CTDE when considering agent $j$ as:

$$Q(\boldsymbol{s}, \boldsymbol{a}) \sim Q(\boldsymbol{s}, a_j, \bar{a}_j), \quad \bar{a}_j = \frac{1}{|\mathcal{N}^j|} \sum_{k \in \mathcal{N}^j} a_k, \tag{7}$$

where $\mathcal{N}^j$ denotes the neighboring agents of agent $j$. However, such an unweighted average neglects the fact that the strengths of agent-agent interactions vary among different agent pairs and over time. Therefore, intuitively, we can improve the approximation into a weighted average to maintain such varying strengths. Moreover, besides reducing the dimension of $\boldsymbol{a}$, we can reduce the dimension of $\boldsymbol{s}$ with a similar technique, further reducing the computational complexity. Generally, we propose the following approximation of the value function of CTDE when considering agent $j$:

$$Q(\boldsymbol{s}, \boldsymbol{a}) \sim Q(s_j, \tilde{s}_j, a_j, \tilde{a}_j), \tilde{a}_j = \frac{1}{W_j} \sum_{k \in \mathcal{N}^j} w^{jk} a_k, \tilde{s}_j = \frac{1}{W_j} \sum_{k \in \mathcal{N}^j} w^{jk} s_k, W_j = \sum_{k \in \mathcal{N}^j} w^{jk}, \tag{8}$$

where $w^{jk}$ is the weight between agent $j$ and $k$, reflecting the strength of interaction between them. On the other hand, in order to make it able to apply our method to PG algorithms with actor-critic structures, we also design a similar weighted average approximation to reduce the computational complexity of policy function, i.e., the actor, as follow:

$$\pi(\boldsymbol{s}) \sim \pi(s_j, \hat{s}_j), \quad \hat{s}_j = \frac{1}{U_j} \sum_{k \in \mathcal{N}^j} u^{jk} s_k, \quad U_j = \sum_{k \in \mathcal{N}^j} u^{jk}, \tag{9}$$

where $u^{jk}$ is the corresponding weight between agent $j$ and $k$.

Next, we will mathematically prove why this approximation holds in Section 4.2 and illustrate how we automatically capture the weights $[w^{jk}]$ and $[u^{jk}]$ and implement such approximation in a practical MARL training algorithm in Section 4.3.

## 4.2 MATHEMATICAL PROOF

Here we prove why the intuitive approximation of the weighted average on the value function holds.

**Theory 1** (Weighted MF Approximation). *When considering agent $j$, the centralized value function $Q(\boldsymbol{s}, \boldsymbol{a})$ can be approximated by $Q(s_j, \tilde{s}_j, a_j, \tilde{a}_j)$.*

*Proof.* Since

$$\tilde{a}_j = \frac{1}{W_j} \sum_{k \in \mathcal{N}^j} w^{jk} a_k, W_j = \sum_{k \in \mathcal{N}^j} w^{jk}, \tag{10}$$

we regard each $a^k, k \in \mathcal{N}^j$ to be the sum of $\tilde{a}_j$ and a small fluctuation as:

$$a_k = \tilde{a}_j + \delta a_{jk}, \quad \frac{1}{W_j} \sum_{k \in \mathcal{N}^j} w^{jk} \delta a_{jk} = \frac{1}{W_j} \sum_{k \in \mathcal{N}^j} w^{jk} (a_k - \tilde{a}_j) = \tilde{a}_j - \tilde{a}_j = 0, \tag{11}$$

and similarly, we have:

$$s_k = \tilde{s}_j + \delta s_{jk}, \quad \frac{1}{W_j} \sum_{k \in \mathcal{N}^j} w^{jk} \delta s_{jk} = 0. \tag{12}$$

Following the expansion in the MF study (Yang et al., 2018), we have:

$$Q(\boldsymbol{s}, \boldsymbol{a}) = \frac{1}{W_j} \sum_{k \in \mathcal{N}^j} w^{jk} Q(s_j, s_k, a_j, a_k) = \frac{1}{W_j} \sum_{k \in \mathcal{N}^j} w^{jk} Q(s_j, \tilde{s}_j + \delta s_{jk}, a_j, \tilde{a}_j + \delta a_{jk}). \tag{13}$$

We denote $Q(s_j, \tilde{s}_j, a_j, \tilde{a}_j)$ as $Q_0$ and expand each term in the sum according to Taylor's formula:

$$(13) = \frac{1}{W_j} \sum_{k \in \mathcal{N}^j} w^{jk} [Q_0 + \nabla_{\tilde{s}_j} Q_0 \cdot \delta s_{jk} + \nabla_{\tilde{a}_j} Q_0 \cdot \delta a_{jk} + o^{jk}]$$

$$= Q_0 + \nabla_{\tilde{s}_j} Q_0 \cdot \frac{1}{W_j} \sum_{k \in \mathcal{N}^j} w^{jk} \delta s_{jk} + \nabla_{\tilde{a}_j} Q_0 \cdot \frac{1}{W_j} \sum_{k \in \mathcal{N}^j} w^{jk} \delta a_{jk} + \frac{1}{W_j} \sum_{k \in \mathcal{N}^j} w^{jk} o_{jk}$$

$$= Q_0 + 0 + 0 + \frac{1}{W_j} \sum_{k \in \mathcal{N}^j} w^{jk} o_{jk} \approx Q_0 \triangleq Q(s_j, \tilde{s}_j, a_j, \tilde{a}_j).$$

$$\tag{14}$$

Hence, the centralized value function $Q(s, a)$ can be approximated with $Q(s_j, \tilde{s}_j, a_j, \tilde{a}_j)$ where the error is within second order small terms.

$\square$

## 4.3 Implementation Details

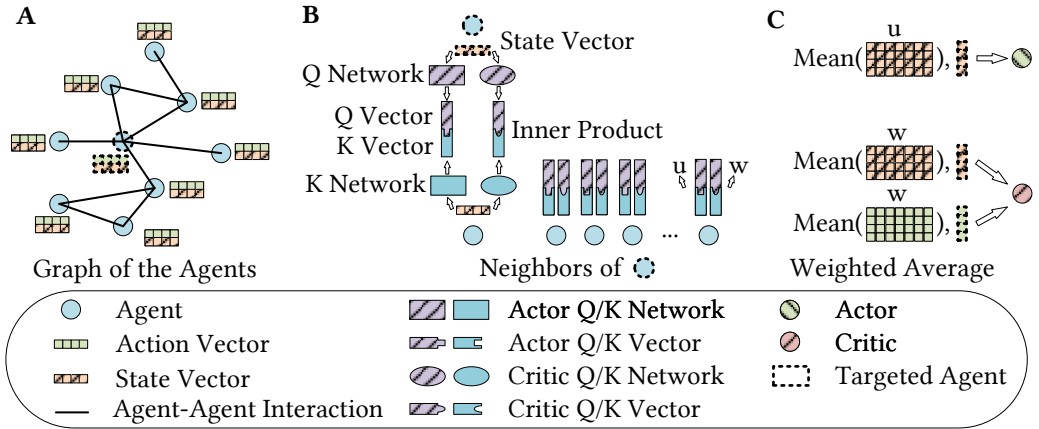

Figure 2: Implementation details of the GAT-MF method. **(A)** Modelling the adjacency among the agents into a graph, where each agent has its state vector and action vector. **(B)** Process of calculating the weights $[w^{jk}]$, $[u^{jk}]$ through graph attention. **(C)** Using the obtained weights to calculate the weighted MF vectors to be the inputs of the actor and critic.

Having proved the correctness of weighted MF approximation, we show how to implement such theory into a practical MARL algorithm in Figure 2. We mainly consider scenarios where the agents have fixed relative positions and therefore we can model the adjacency among them into a graph $\mathcal{G}$ (Figure 2A). In order to obtain the weights $[w^{jk}]$ and $[u^{jk}]$, which can reflect the varying strengths of interactions among each agent and its neighbors, we design a variant of graph attention (GAT) mechanism to automatically learn them.

In detail, each agent is assigned with a pair of query (Q) networks $\mathcal{Q}_a^j, \mathcal{Q}_c^j$ and key (K) networks $\mathcal{K}_a^j, \mathcal{K}_c^j$ with learnable parameters, which respectively correspond to the actor and critic. In each time step, $\mathcal{Q}_a^j, \mathcal{Q}_c^j, \mathcal{K}_a^j, \mathcal{K}_c^j$ take in the current local state vector $s_j$ and produce a pair of Q vectors $q_a^j, q_c^j$ and a pair of K vectors $k_a^j, k_c^j$, which all have the same dimension. Then, we calculate the weights through the inner product of the corresponding Q and K vectors (Figure 2B), as:

$$u^{jk} = (q_a^j)^T k_a^k, \quad w^{jk} = (q_c^j)^T k_c^k. \tag{15}$$

By this mechanism, we obtain the agent-pair-specific and time-varying weights, reflecting the strengths of agent-agent interactions. Finally, we weigh the state and action vectors by the weights and calculate the MF vectors, which are then used as the inputs of the actor and critic (Figure 2C). The parameters of $\mathcal{Q}_a^j, \mathcal{Q}_c^j, \mathcal{K}_a^j, \mathcal{K}_c^j$ are updated along with the actor and critic networks, and we also apply the technique of target network with delayed parameters synchronization to them.

We combine our GAT-MF method with the original MADDPG algorithm and show the step-by-step training algorithm in Appendix A. Considering the homogeneity among the agents, the network parameters are shared among the agents, reducing memory consumption. We will first evaluate our method in a grid-world task with 100 agents and provide some straightforward visualizations of the learned policies in Section 5. We will then extend the experiment into a real-world task with more than 3000 agents in Section 6, fully verifying the ability of our method to scale up to scenarios with a very large number of agents. Our experiments are implemented with PyTorch and the source codes and data for reproducibility are posted into supplementary materials and also at an anonymous repository https://github.com/MyGithubForReview/Large-Scale-MARL.

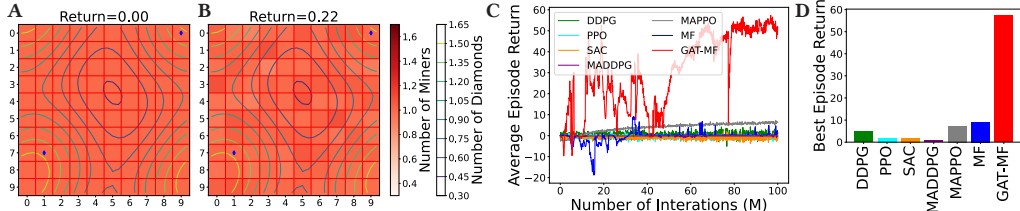

Figure 3: Experimental settings and results in the grid world. **(A)** An initialized example of the grid world with randomly located diamonds and uniformly distributed miners. **(B)** The return after 10 steps of random actions. **(C)** Training curves of the proposed GAT-MF, the baselines, and the ablation study. **(D)** Best performance of each method.

# 5 EXPERIMENT 1: GRID-WORLD TASK

## 5.1 EXPERIMENTAL SETTINGS

First, we start with a diamonds-seeking task in a manual grid world consisting of $10 \times 10$ grids with loop boundaries and the grids correspond to 100 agents in total. We show a randomly initialized example of the grid word in Figure 3A. Initially, there are two diamond veins buried in two random grids (blue markers in the figure) and the density of diamonds over the world, which is denoted by the contour lines, follows Gaussian distributions centered at the two grids. There are also miners uniformly distributed over the grids, whose number is denoted by the background color. The total number of obtained diamonds is calculated by summing up the product of the number of diamonds and miners at each grid and the return is defined by how many more diamonds are obtained than initialization. In each time step, 10% of the miners from each grid move to one of its neighbors, and the task of the agents is to give the exact numbers of miners to each of the four neighbors to efficiently redistribute the miners and get as many diamonds as possible. We show the distribution of miners and the return after 10 steps of random movements in Figure 3B as an example.

## 5.2 PERFORMANCE EVALUATION

As we mentioned in Section 4.3, we mainly combine our proposed GAT-MF with the MADDPG algorithm and train the model. We set the local state $s_j$ at grid $j$ to be a five-dimensional vector consisting of the densities of diamonds at the center and the four corners of the grid. The local one-step reward $r$ is set to be the difference in the number of obtained diamonds by the grid and its neighbors before and after the step. During the training, we set 10 steps to be one episode and randomly reinitialize the grid world after each episode. We compare the performance of our method with various baseline RL algorithms, which are listed as follows:

- **Global agent methods**: In this group of baselines, we apply widely used single-agent RL methods including **DDPG** (Silver et al., 2014), **PPO** (Schulman et al., 2017) and **SAC** (Haarnoja et al., 2018) to the task. We consider the single-agent to be a global one, which takes in the global state $s$ from all grids and gives the joint action $a$ of all grids in the form of a concatenated high dimensional vector.

- **CTDE MARL methods**: In this group of baselines, we apply original versions of **MADDPG** (Lowe et al., 2017) and **MAPPO** (Yu et al., 2021), which are popular CTDE methods.

Specially, we further design the following ablation study to verify the vital role of our GAT design:

- **Ablation study, w/o GAT**: We substitute the proposed GAT-MF with the unweighted **MF** method as we described in Section 4.1 while keeping the rest parts identical.

We show the training curves over 100M agent-environment interactions in Figure 3C and extract the best performance of each method in Figure 3D. The results illustrate the superior performance of our method with a large margin. The ablation version of our method, i.e., the **MF**, ranks second, proving the validity of the mean field approximation and the large improvement of our method comparing it

verifies the key role of our GAT design. Though MAPPO, one of the SOTA MARL methods, ranks third, all global agent methods and CTDE MARL methods perform badly due to the high complexity caused by the high dimensional vectors. This again stresses the necessity of using the mean-field approximation to reduce the dimension when the number of agents is large.

## 5.3 VISUALIZATIONS OF THE LEARNED POLICY

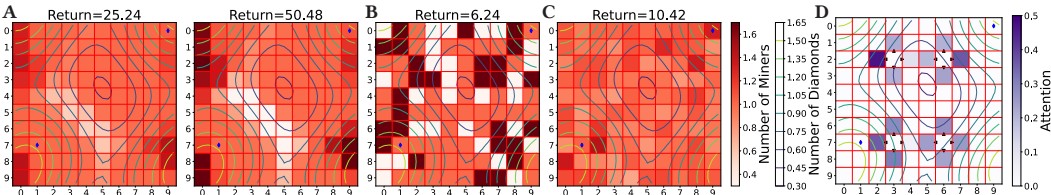

Figure 4: Visualizations of the learned policies. **(A)** The distribution of miners after applying the policy from a trained GAT-MF model for 5 and 10 steps. **(B)** The distribution of miners after applying the policy from a trained DDPG model for 10 steps. **(C)** The distribution of miners after applying the policy from a trained MAPPO model for 10 steps. **(D)** The learned attentions of some actors at the first step from a trained GAT-MF model.

To help understand what the agents learned through training with our method, here we provide some visualizations of the learned policies. In Figure 4A, we show the distribution of miners after applying the policy from a trained GAT-MF model for 5 and 10 steps. From these we find the agents have learned reasonable policy, quickly making the miners gather at grids with the most diamonds, e.g., the left-upper corner in the shown example. On the other hand, we show in Figure 4B & C the distribution of miners after applying policies from a trained DDPG model and a trained MAPPO model for 10 steps, which are respective representatives of global agent methods and CTDE MARL methods. The DDPG policy tends to be like random movements and the MAPPO policy tends to only do very slight movements. We show more visualizations of policies from models trained via other baseline methods in Appendix C and find none of them help the agents learn efficient policy. Once again, we verify the advantage of our proposed method over the existing ones.

We also visualize the learned attentions of some actors at the first step from a trained GAT-MF model in Figure 4D. We find the attentions vary among the agents and reasonably, the agents tend to pay larger attention to the neighboring agents which are in grids with more diamonds. We show more visualizations of the attentions in Appendix C and from all these results, we can verify the validity of the GAT design in capturing the varying strengths of agent-agent interactions.

## 6 EXPERIMENT 2: REAL-WORLD METROPOLITAN TASK

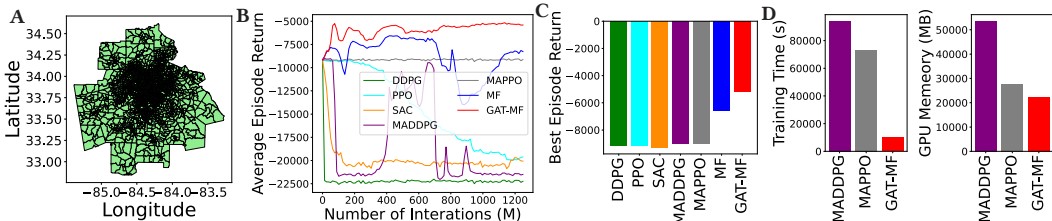

Figure 5: Experimental settings and results in the real world. **(A)** The map of Atlanta, where each block is one unit in the vaccine-allocation task and corresponds to one agent. **(B)** Training curves of the proposed GAT-MF, the baselines, and the ablation study. **(C)** Best performance of each method. **(D)** Computational efficiency comparison.

### 6.1 EXPERIMENTAL SETTINGS

In order to verify the ability of our method to scale up to scenarios with a larger number of agents, we conduct another experiment on the real-world task of COVID-19 vaccine allocation. During the pandemic of COVID-19, people in metropolis with high population density and frequent population mobility suffer a high risk of infection (Stier et al., 2020). However, vaccines, one of the most important medical resources, are in shortage during the early stage of COVID-19. Therefore, in order to minimize the overall infections, finding an efficient strategy to allocate the limited vaccines to the key population group, e.g., the elders or the health workers, in the metropolis is of vital importance. Previous researchers have built a precise simulator with real-world data to infer the total infections with different vaccine-allocation strategies (Chen et al., 2022). We conduct our experiments based on the simulator in Atlanta, where there are 3130 blocks, corresponding to 3130 agents (Figure 5A). People in each block have different age structures, time-varying moving and contacting patterns, and thus have complex time-varying risks of infection. The task of the agents is to find an efficient way to allocate the limited vaccines among the blocks at each time step, reduce the local risk of infection in certain key blocks, and thus minimize the overall infections, which is opposite to the return. We show a more detailed background of this experiment in Appendix D.

### 6.2 PERFORMANCE AND COMPUTATIONAL EFFICIENCY

During the training, we set 24 hours as one step and an episode consists of 63 steps (9 weeks). The local state $s_j$ at block $j$ consists of the current number of susceptible, infected, and dead people in this block and we set the one-step reward $r$ to be opposite to the number of newly infected people in the step. We compare the performance of our method with the same baselines and ablation study in Section 5.2, setting the available number of vaccines identical among our method and the baselines.

We show the smoothed training curves over more than 1.2B agent-environment interactions in Figure 5B and summarize the best performance of each method in Figure 5C. The results illustrate that in this larger scenario with thousands of agents, all the baseline methods failed to learn useful strategies and their performance is even worse than a randomly initialized model. In contrast, our method reaches a good performance rapidly and converges stably, proving its superior performance again. On the other hand, the unweighted MF method still ranks second, which again verifies the validity of the mean field approximation and the key role of our GAT design.

We compare the computational efficiency[1] of our method with the two CTDE MARL methods in Figure 5D. The results show that with the GAT-MF design, which greatly reduces the input dimension of the actor and critic, our method is $9.4\times$ and $7.3\times$ faster than MADDPG and MAPPO, training with the same number of agent-environment interactions. Besides, our method takes only 41.5% and 80.8% of GPU memory comparing MADDPG and MAPPO. In a nutshell, our method works especially well in this scenario where there is a much larger number of agents, not only reaching superior performance but also greatly reducing the training time and GPU memory consumption.

## 7 CONCLUSIONS

In this paper, we aimed at scaling up the number of agents in MARL algorithms and proposed the GAT-MF method. We first intuitively analyzed its advantages over the existing CTDE methods and the unweighted MF method and then mathematically proved its correctness. We implemented the mathematical theory into a practical MARL algorithm through the graph attention mechanism and conducted extensive experiments in both manual and real-world scenarios with up to more than 3000 agents. All the results indicated the superior performance and high computational efficiency of our method. Because of the advantages of our method, it has great potential to be applied in solving various real-world large-scale problems.

One main limitation of our method is that the GAT mechanism requires the agents to have fixed relative positions and thus the adjacency among them can be modeled as a static graph. However, in some scenarios, the agents keep moving and one possible solution for this is to extend the static graph in this paper into a dynamic one and modify the GAT mechanism to work on dynamic graphs. We leave this as a potential direction for future work.

---

[1]Test on NVIDIA A100 GPU

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

# A   TRAINING ALGORITHM

Here we show the step-by-step training algorithm in Algorithm 1.

---

**Algorithm 1** Multi-agent Training with GAT-MF

---

**Require:** Number of agents $n$, number of training episodes $M$, model update interval $I$, reward discount factor $\gamma$

**Ensure:** The trained multi-agent model

1: Initialize actor networks $\pi$, initialize query (Q) networks $\mathcal{Q}_a, \mathcal{Q}_c$ and key (K) networks $\mathcal{K}_a, \mathcal{K}_c$, the parameters are shared among the agents

2: Initialize the critic network $Q$

3: Copy $\pi, \mathcal{Q}_a, \mathcal{Q}_c, \mathcal{K}_a, \mathcal{K}_c$ and get the corresponding target networks $\dot{\pi}, \dot{\mathcal{Q}}_a, \dot{\mathcal{Q}}_c, \dot{\mathcal{K}}_a, \dot{\mathcal{K}}_c$

4: Copy $Q$ and get the corresponding target network $\dot{Q}$

5: Initialize the experience replay buffer $\mathcal{B}$

6: **for** episode = 1 to $M$ **do**

7:     Initialize a random process $\mathcal{N}$ for action exploration

8:     Initialize the environment and obtain the initial state $\boldsymbol{s} = (s_1, ..., s_n) \in \mathcal{S}$

9:     **for** $t$ = 1 to max-episode-length **do**

10:         Calculate the Q vectors and K vectors of each agent:
$$q_a^j = \mathcal{Q}_a(s_j), \quad q_c^j = \mathcal{Q}_c(s_j), \quad k_a^j = \mathcal{K}_a(s_j), \quad k_c^j = \mathcal{K}_c(s_j)$$

11:         **for** $j$ = 1 to $n$ **do**

12:            Calculate the **GAT** weights for agent $j$ through the Q vectors and K vectors:
$$u^{jk} = (q_a^j)^T k_a^k, \quad k \in \mathcal{N}^j$$

13:            Calculate the local action with noise through **GAT-MF**:
$$a_j = \pi(s_j, \hat{s}_j) + \mathcal{N}, \quad \hat{s}_j = \frac{1}{U_j} \sum_{k \in \mathcal{N}^j} u^{jk} s_k, \quad U_j = \sum_{k \in \mathcal{N}^j} u^{jk}$$

14:         **end for**

15:         Execute the joint action $\boldsymbol{a} = (a_1, ..., a_n) \in \mathcal{A}$ in the environment, observe the next state $\boldsymbol{s}' = (s_1', ..., s_n') \in \mathcal{S}$ and the local one-step rewards $\boldsymbol{r} = (r_1, ..., r_n) \in \mathbb{R}^n$

16:         Store $(\boldsymbol{s}, \boldsymbol{a}, \boldsymbol{r}, \boldsymbol{s}')$ into $\mathcal{B}$

17:         **if** reach the update interval $I$ **then**

18:            Sample a random batch of experiences $(\boldsymbol{s}, \boldsymbol{a}, \boldsymbol{r}, \boldsymbol{s}')$ from $\mathcal{B}$

19:            **for** j = 1 to $n$ **do**

20:                Set $y_j = r_j + \gamma \dot{Q}(s_j', \tilde{s}_j', \dot{a}_j^{s'}, \tilde{a}_j^{s'})$ where $\dot{a}_j^{s'} = \dot{\pi}(s_j', \hat{s}_j'), \dot{\boldsymbol{a}}^{s'} = (\dot{a}_1^{s'}, ..., \dot{a}_n^{s'})$

                        $\hat{s}_j'$ is the **GAT-MF** vector of $\boldsymbol{s}'$ with weights from $\dot{q}_a'^j = \dot{\mathcal{Q}}_a(s_j')$ and $\dot{k}_a'^{\mathcal{N}^j} = \dot{\mathcal{K}}_a(s_{\mathcal{N}^j}')$

                        $\tilde{s}_j'$ is the **GAT-MF** vector of $\boldsymbol{s}'$ with weights from $\dot{q}_c'^j = \dot{\mathcal{Q}}_c(s_j')$ and $\dot{k}_c'^{\mathcal{N}^j} = \dot{\mathcal{K}}_c(s_{\mathcal{N}^j}')$

                        $\tilde{a}_j^{s'}$ is the **GAT-MF** vector of $\dot{\boldsymbol{a}}^{s'}$ with weights from $\dot{q}_c'^j$ and $\dot{k}_c'^{\mathcal{N}^j}$

21:                Calculate $L_j = (y_j - Q(s_j, \tilde{s}_j, a_j, \tilde{a}_j))^2$

                        $\tilde{s}_j$ is the **GAT-MF** vector of $\boldsymbol{s}$ with weights from $q_c^j = \mathcal{Q}_c(s_j)$ and $k_c^{\mathcal{N}^j} = \mathcal{K}_c(s_{\mathcal{N}^j})$

                        $\tilde{a}_j$ is the **GAT-MF** vector of $\boldsymbol{a}$ with weights from $q_c^j$ and $k_c^{\mathcal{N}^j}$

22:                Calculate $J_j = Q(s_j, \tilde{s}_j, a_j^s, \tilde{a}_j^s)$ where $a_j^s = \pi(s_j, \hat{s}_j), \boldsymbol{a}^s = (a_1^s, ..., a_n^s)$

                        $\hat{s}_j$ is the **GAT-MF** vector of $\boldsymbol{s}$ with weights from $q_a^j = \mathcal{Q}_a(s_j)$ and $k_a^{\mathcal{N}^j} = \mathcal{K}_a(s_{\mathcal{N}^j})$

                        $\tilde{a}_j^s$ is the **GAT-MF** vector of $\boldsymbol{a}^s$ with weights from $q_c^j = \mathcal{Q}_c(s_j)$ and $k_c^{\mathcal{N}^j} = \mathcal{K}_c(s_{\mathcal{N}^j})$

23:            **end for**

24:            Update parameters of $Q, \mathcal{Q}_c, \mathcal{K}_c$, minimizing $L = \sum_j L_j$

25:            Update parameters of $\pi, \mathcal{Q}_a, \mathcal{K}_a$, maximizing $J = \sum_j J_j$

26:            Update $\dot{\pi}, \dot{\mathcal{Q}}_a, \dot{\mathcal{Q}}_c, \dot{\mathcal{K}}_a, \dot{\mathcal{K}}_c$ from parameters of $\pi, \mathcal{Q}_a, \mathcal{Q}_c, \mathcal{K}_a, \mathcal{K}_c$ via soft replace

27:            Update $\dot{Q}$ from parameters of $Q$ via soft replace

28:         **end if**

29:     **end for**

30: **end for**

31: **return** The train models for execution $\pi, \mathcal{Q}_a, \mathcal{K}_a$

---

## B NUMERICAL RESULTS FOR THE BAR PLOTS

In this section, we provide numerical results of the bar plots in Figure 3 and Figure 5.

The numerical results of the bar plot in Figure 3D are shown in Table 1.

Table 1: Numerical results of the bar plot in Figure 3D

| Group | Method | Best Episode Return |
|---|---|---|
| Global agent methods | DDPG | 4.827 |
| | PPO | 1.846 |
| | SAC | 1.934 |
| CTDE MARL methods | MADDPG | 0.841 |
| | MAPPO | 7.359 |
| Ablation study | MF | 9.156 |
| **Our method** | **GAT-MF** | **57.370** |

The numerical results of the bar plot in Figure 5C are shown in Table 2.

Table 2: Numerical results of the bar plot in Figure 5C

| Group | Method | Best Episode Return |
|---|---|---|
| Global agent methods | DDPG | -9127.401 |
| | PPO | -9185.278 |
| | SAC | -9293.688 |
| CTDE MARL methods | MADDPG | -9023.012 |
| | MAPPO | -9013.738 |
| Ablation study | **MF** | -6572.228 |
| **Our method** | **GAT-MF** | **-5162.126** |

The numerical results of the bar plots in Figure 5D are shown in Table 3.

Table 3: Numerical results of the bar plot in Figure 5D

| Method | Training Time (s) | GPU Memory (MB) |
|---|---|---|
| MADDPG | 94041 | 53553 |
| MAPPO | 73208 | 27687 |
| **GAT-MF** | **9980** | **22379** |

## C    VISUALIZATIONS OF THE LEARNED POLICY IN EXPERIMENT 1

In this section, we provide more visualizations of the learned policy with different methods in experiment 1. We show the distribution of miners after applying the policy from a trained GAT-MF model, a trained ablation model, a trained MAPPO model, a trained MADDPG model, a trained SAC model, a trained PPO model, and a trained DDPG model for 1, 3, 5, 8, and 10 steps in Figure 6. We also visualize the learned attentions of some more actors at the first step from a trained GAT-MF model in Figure 7.

## D    DETAILED BACKGROUND OF EXPERIMENT 2

In this section, we introduce more details of the background of the real-world vaccines-allocation experiment. The long-lasting global pandemic spreading of COVID-19 has caused countless damage to the social economy and people's daily life. With the aim of better understanding the pandemic spreading dynamics, there exist many works on modeling and simulating the COVID-19 pandemic spreading with various mathematical models and data sources (Chang et al., 2021a;b; Chen et al., 2022). In this paper, we mainly conduct our experiments based on the Behavior and Demography informed epidemic model (BD model) (Chen et al., 2022), which is an improved version of the meta-population model (Chang et al., 2021a).

In general, the BD model simulates the pandemic spreading inside the metropolitan statistical areas (MSA[2]s), with the smallest unit of census block groups (CBG[3]s) and points of interest (POI[4]s). Typically, there are thousands of CBGs and more than ten thousand of POIs within one MSA. The BD model divides the pandemic spreading process into two parts, i.e., Intra-CBG transmissions Inter-CBG transmissions, and models them respectively. For the Intra-CBG transmissions, it maintains a local Susceptible-Exposed-Infected-Removed (SEIR) model in each CBG, where S denotes susceptible, E denotes exposed, I denotes infected and R denotes removed. There is a proportion of infected cases become reported cases and the proportion is decided by the testing capability. Only reported cases are observable while the rest of the infected cases are not observable. There are a proportion of removed cases turn out to be deaths according to the infection-fatality rate (IFR) in each CBG while others turn out to recover. As is known to us, the death rate is strongly related to age and thus the IFR is estimated by the population age structure of each CBG and age-specified death risks. On the other hand, the BD-model mainly models the Inter-CBG transmissions part based on the population mobility and contact network among CBGs and POIs. Inter-CBG transmissions happen when susceptible individuals from a certain CBG encounter infected individuals from another CBG when visiting a POI. The transmission probability varies among the POIs, which is positively related to people's average dwelling time at a certain POI and is inversely proportional to the POI's floor space, reflecting the population density in such POI. The effect of vaccines on the transmission process is modeled as an equivalent reduction in the infection rate, which means the probability for susceptible individuals in a certain CBG to be infected and turn into an exposed case in contact with infected individuals reduces proportionally to the percentage of vaccinated individuals in such CBG. We assume that an individual will obtain 100% immunity after one dose of vaccine injection for simplification in this paper while the real-world situations of two-injection vaccines and $< 100\%$ immunity can be modeled with no essential difference by simply changing the parameters.

The required data of population mobility, i.e., how people from each CBG visit each POI and encounter people from other CBGs, are originally captured by previous researchers from the Safe-Graph open data[5] and are available online[6] (Chang et al., 2021a). In the study of BD-model, the researchers further process the data into a suitable form to feed into the simulator. They have verified the accuracy of the BD model by comparing the simulation results by computers with the reported real-world situations. Results show that the model can reflect the real-world pandemic spreading situations precisely and thus we can conduct experiments based on it with confidence.

---

[2]Regions with a relatively high population density and close economic ties throughout the area.

[3]The smallest geographical unit for which the bureau publishes sample data.

[4]Specific locations that someone may find useful or interesting.

[5]https://www.safegraph.com/

[6]https://covid-mobility.stanford.edu//datasets/

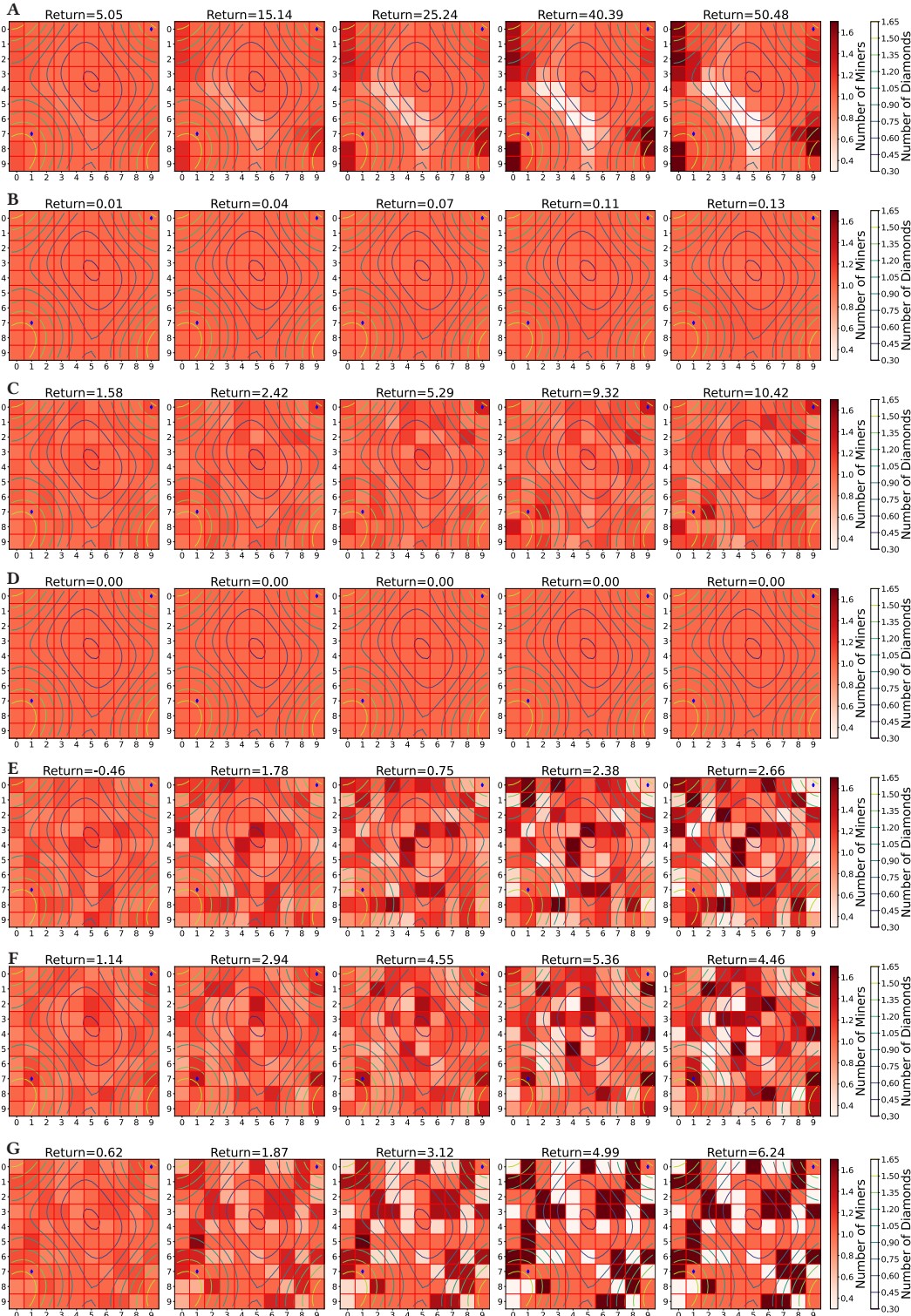

Figure 6: More visualizations of the learned policies. **(A)-(G)** The distribution of miners after applying the policy from a trained GAT-MF model, a trained ablation model, a trained MAPPO model, a trained MADDPG model, a trained SAC model, a trained PPO model, and a trained DDPG model for 1, 3, 5, 8, and 10 steps, from left-top to right-bottom, correspondingly.

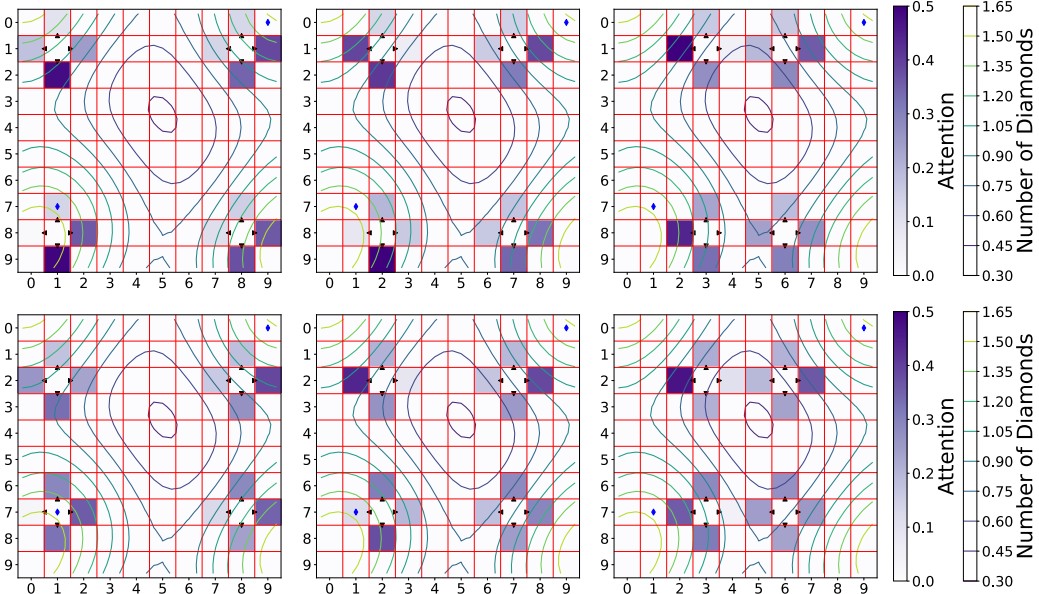

Figure 7: More visualizations of the learned policies. The learned attentions of some more actors at the first step from a trained GAT-MF model. Since one panel cannot hold the attentions of all the actors, we show the attentions of four actors in each panel.

