# OpenReview forum: "Very Large Scale Multi-Agent Reinforcement Learning with Graph Attention Mean Field"
_ICLR.cc/2023/Conference — Submitted to ICLR 2023_

### Official Review · Reviewer_cg8c · 2022-10-24

**Confidence:** 3
**Correctness:** 2
**Technical Novelty And Significance:** 3
**Empirical Novelty And Significance:** 2
**Recommendation:** 3

**Clarity, Quality, Novelty And Reproducibility:**

The paper is rather clear, though I could not answer some issues (see below). The novelty is somewhat limited; the paper is a rather straightforward extension of the mean-field method. I have a major concern about reproducibility. The experiments are presented using one seed only (?) and are rather unstable, which raises the question if the conclusions are statistically significant.

**Strength And Weaknesses:**

Strength:
- an interesting and important problem
- natural approach

Weaknesses
- poor empirical evolution

**Summary Of The Paper:**

The paper develops a multi-agent algorithm suitable for handling a large number of agents by means of the mean-field approximation.

**Summary Of The Review:**

The paper develops a multi-agent algorithm suitable for handling a large number of agents by means of the mean-field approximation. The major contribution is utilising an attention mechanism to quality the strength of interactions. This is a very natural approach and likely to be beneficial.

My major concern is about empirical evaluation. The grid world environment, as far as I understood, does not include any form of interaction. Doesn't it factorise into $100$ of completely independent environments?

Questions and comments:
 - Is the methods for cooperative/mixed/competitive MARL?
 - I do not quite understand the statement of 'Theory 1'. What does it mean 'approximated'?
 - I am not sure what $\mathcal{N}_j$  are used in the experiments. Can the authors clarify that?
 - The graph on Fig 3 is hard to read. Also, "Best episode return" is a somewhat unusual metric.
 - The method seems not very stable. It is not unusual for MARL trainings; however, the paper would benefit from some discussion or more experiments with this regard.

---

### Official Review · Reviewer_vFJW · 2022-10-24

**Confidence:** 3
**Correctness:** 2
**Technical Novelty And Significance:** 2
**Empirical Novelty And Significance:** 2
**Recommendation:** 5

**Clarity, Quality, Novelty And Reproducibility:**

Even though the beginning of the paper and the numerical test cases are quite clearly presented, I feel that some improvements are needed due to the gap in the literature review (which makes it hard to know to what extent the method is new) and the theoretical result (claimed as one of the main contributions).

**Strength And Weaknesses:**

Strength: (1) The beginning of the paper (motivations and part of the literature review) is quite clear. (2) The experiments show that the method can be used for a relatively large number of agents (a few thousands).

Weaknesses:
(1) References: It seems that there are already several papers on multi-agent problems using graph attention mechanism, such as (to cite just one example but I do not know if this is the most relevant one):
Multi-Agent Game Abstraction via Graph Attention Neural Network, Liu et al., AAAI’20
Furthermore, there is a growing literature on mean-field reinforcement learning or reinforcement learning for mean field games, whose goal is precisely to tackle games with a very large number of players. Given the similarities in key-words and goals, it could be relevant to mention the existence of this literature and the main similarities or differences.

(2) Although (Yang et al., 2018) is very related to your work, I would recommend clarifying that the notion of solution is quite different. They consider a Nash equilibrium whereas you focus on a joint optimization problem.

(3) “Theory 1” (“Theorem 1”?): In its current form, it is hard to get any insight from this result or to check its correctness. Indeed, any function “can be approximated” (to some degree) by any other function. Could you please make it more precise? It would be nice to provide a quantitative statement, and to spell out clearly the assumptions (e.g., on the Q function).

Typos and minor questions:
Page 3 and 4: In the model, what are the assumptions on the state and action space? Are they finite? If not, are they compact?
Page 4: “possibility distribution” → “probability distribution”, “collection of possibility distribution” → “collection of possibility distributions”
Page 5: “Theory 1” → “Theorem 1”
Page 6: formula (15): Why are these “time-varying weights”?
Page 7: “the rest parts” → “the other parts”
Page 8: Figure 4: Is it correct that the DDPG method yields a higher reward than the GAT-MF method? Could you comment on this please?
Page 9: Section 6.2: How is computed the “1.2B” in the expression “1.2B agent-environment interactions”?


**Summary Of The Paper:**

This paper studies multi-agent problems with a very large number of agents. In the past, such problems have been tackled using a combination of reinforcement learning and mean field approximations. This approach is valid provided the interactions between the players are homogeneous in the sense that every player interacts roughly in the same way with all the other players. Here, the authors consider a problem in which the interactions have some network structure, meaning that each agent may interact more with some agents rather than some others. To address this aspect, they propose to use graph attention networks. They first explain how to rewrite the Q-function using local information, then they design a graph attention mechanism and finally they conduct experiments on two examples, with up to thousands of agents.

**Summary Of The Review:**

I would recommend clarifying the comparison with the existing literature (so that we understand better the algorithmic contribution) and the theoretical contribution.

---

### Official Review · Reviewer_T6mP · 2022-10-25

**Confidence:** 4
**Correctness:** 2
**Technical Novelty And Significance:** 2
**Empirical Novelty And Significance:** 2
**Recommendation:** 3

**Clarity, Quality, Novelty And Reproducibility:**

The paper is clear. The paper doesn’t seem very novel (please refer to the previous section). Code is provided.


**Strength And Weaknesses:**

### Strengths

The paper is clearly written and the idea of modeling the strength of interaction between agents with an attention mechanism is interesting. The two environments used are

### Weaknesses

I don’t think it is fair to list the result in Section 4.2 (Weighted MF Approximation) as one of the main contributions of this paper. This is just a minor modification of the result from Yang et al., 2018 that includes the weights and whose proof follows trivially from the original result.

The paper is not well-positioned in the literature. There are many missing references to relevant papers. This makes it very hard to properly assess the novelty of this work. A few of these papers are listed below:

 * Guestrin et al., 2002. Coordinated Reinforcement Learning.
 * Nair et al., 2005. Networked Distributed POMDPs: A Synthesis of Distributed Constraint Optimization and POMDPs.
 * Peter Sunehag et al., 2018. Value-Decomposition networks for cooperative multi-agent learning based on team reward.
 * Bohmer et al., 2020. Deep Coordination Graphs.
 *Oliehoek et al., 2021. A sufficient Statistic for Influence in Structured Multiagent Environments.

If I’m not mistaken the critics in MADDPG and MAPPO take as input the global state and the actions of all the other agents. If this is the case, the experiments are missing two baselines that I believe the method should be compared against:
 * A policy and value function that condition only on the pivot agent’s local states such as independent q-learning (Tan, 1993). This is meant to show that the neighbors' information is important and that the weighted mean-field approximation is really needed.
 * A policy and value function that condition on the pivot agent’s local states, and the neighbor agents’ actions and local states. This is meant to show that feeding the weighted mean-field approximation is more effective than just feeding the neighbors’ local states and actions.

 (Tan, 1993) MultiAgent Reinforcement Learning Independent vs Cooperative Agents.

**Other questions/suggestions**

Is the GAT learned end-to-end using the reward signal?

As far as I can see, the two environments are Dec-POMDPs since the agents’ receive only their own local states and the neighbors’ local states. In general, these are not Markovian. Don’t you need to condition the policy and value function on the action-observation history? Also, why does the policy condition on the neighbors’ local states but not their actions?

I didn’t find any information about the experimental setup: number of random seeds, hyperparameter configurations… Are the results reported in Figures 3 and 5 averaged over different trials?

The paper states that the method requires the agents to have fixed relative positions. Is this the case in the grid-world task? Aren’t the miners moving independently?

**Minor mistakes/typos**

Specify how the actions are represented. Are they one-hot encoding vectors?

I believe the agent’s subscript is missing in equations (8) and (9). $Q_j(s,a)$ and $\pi_j(s)$.

In Section 5.1 when you say “Gaussian distributions centered at the two grids” do you mean centered at the two “cells” that contain the diamond veins?

In Figure 3. Why do the contours show two peaks at (0,0) and at (9,9)? Aren’t there just two Gaussians centered at the diamond veins?


**Summary Of The Paper:**

This paper proposes to change the mean-field approximation of the Q function proposed by Yang et al., 2018 into a function that takes the weighted average of the neighbors’ actions rather than the simple average.  Weights measure the degree of interaction between an agent and its neighbors. These are computed at every timestep using an attention mechanism so that the weights can adapt to the current context. The method is shown to outperform the baselines in a grid world scenario and a more complex environment that simulates the problem of vaccine allocation during COVID-19.


**Summary Of The Review:**

Overall, I think this paper is not ready for publication. My suggestion to the authors is to relax some of their claims (listing the result in Section 4.2 as a contribution is not appropriate in my opinion), try to position this work better in the literature, and add more baselines to the experiments so as to really demonstrate the benefits of the method.

---

### Decision · Program_Chairs · 2023-01-20

**Decision:**

Reject

**Justification For Why Not Higher Score:**

Unanimous decision by reviewers.

**Justification For Why Not Lower Score:**

N/A

**Metareview: Summary, Strengths And Weaknesses:**

Reviewers unanimously agree that the weaknesses of the paper outweigh its strengths. Reviewers remarked a lack of novelty, an insufficient positioning with respect to existing literature, and voiced concerns regarding the empirical validation. Therefore, I recommend rejection.